# Sugar consumption in schoolchildren from southern Spain and influence on the prevalence of obesity

**Leticia Heras-Gonzalez[1], José Antonio Latorre[2], Manuel Martinez-Bebia[2], Nuria Gimenez-Blasi[3], Fátima Olea-Serrano**  **[1], Miguel Mariscal-Arcas[1]** *

**1** Research Group Nutrition, Diet and Risk Assessment (AGR-255), Department of Nutrition and Food Science, University of Granada, Granada, Spain, **2** Department of Food Technology, Nutrition and Food Science, University of Murcia, Lorca, Murcia, Spain, **3** School of Heath Science, University of Isabel I, Burgos, Spain

* mariscal@ugr.es

**Data Availability Statement:** Data cannot be shared publicly because of protection of personal data. Data are available from the Andalusian Public

## Abstract

### Aim

The main cause of childhood overweight/obesity is an imbalance between energy intake and energy expenditure. The objective was to determine whether the intake by Spanish schoolchildren of sugars from habitually consumed foods and drinks can be related to overweight/obesity.

### Methods

*Subjects* The study included 657 schoolchildren between 7–10 years from educational centers in Southern Spain. These children live under the influence of the Mediterranean diet. *Design* Participants completed an encoded questionnaire with three sections: a) data on sex, age, educational center, school year, and life/family habits, among others; b) semi-quantitative food frequency questionnaire related to the previous 12 months; and c) information on anthropometrics and physical activities.

### Results

Obesity was observed in 10.9% of the children. The daily activity questionnaire showed a mean energy expenditure of 8.73 (1.33) MJ/day. The study considered foods that supply carbohydrates in any form (total carbohydrates, starch, total sugars, added sugars, and free sugars). The likelihood of overweight/obesity was significantly greater with a higher intake/day of total sugars, starch, added sugars, and free sugars. The likelihood of normal weight was significantly greater with lower energy expenditure in sedentary activities (OR = 3.03), higher energy expenditure in sports activities (OR = 1.72), and higher total activity/day measured as METs (OR = 8.31).

### Conclusions

In this population, overweight/obesity was influenced by the physical activity of the children and by their intake of energy, total sugars, starch, added sugars, and free sugars. Further

Health Service Institutional (FIBAO, IBS-Granada, Spain) Data Access / Ethics Committee (contact via Sarah E. Biel Gleeson sbiel@fibao.es) for researchers who meet the criteria for access to confidential data.

**Funding:** This study was supported by the Andalusian Regional Government (Nutrition, Diet and Risks Assessment: AGR255), and European Regional Development Fund (FEDER) and Carlos III Health Institute (ISCIII) - FEDER-ISCIII PI14/01040.

**Competing interests:** No authors have competing interests.

studies are warranted to verify this observation and explore the implications for public health policies.

## 1. Introduction

Childhood obesity is one of the most severe public health challenges of this century. It is associated with an increased risk of the early onset of diabetes, cardiovascular disease, and respiratory problems and a higher likelihood of fractures and hypertension, insulin resistance, and psychological disorders, among other severe health complications [1].

The main cause of childhood overweight and obesity is an imbalance between energy intake and energy expenditure. However, a hyperenergetic diet (e.g., excessive energy consumption) and a lack of physical activity are not the only influential factors. Childhood obesity is also associated with social and economic development, and policies on agriculture, transport, urban planning, environment, education, and food processing, distribution, and commercialization, which all require special attention in the fight against the obesity epidemic. In particular, it should be taken into account that the prevalence of overweight and obesity is higher in children with lower socioeconomic and cultural levels [2,3]. Importantly, overweight/obesity, and related diseases are largely preventable [1].

Humans are genetically programmed to like sweetness [4], which has strong unconscious effects and is the taste most strongly related to pleasure. It is also clear that we often like what we habitually consume and that our palate can be educated, and people prefer different degrees of sweetness or saltiness [5,6]. Free sugars (refined or non-refined) are added to food by manufacturers and/or consumers, including monosaccharides/disaccharides introduced into foods and drinks and also sugars naturally present in honey and in fruit syrups or juices, although the WHO excludes the intrinsic sugars present in whole fresh fruit and vegetables from the classification of free sugars [7]. Sugar should preferably be consumed as part of a main meal and in a natural form (sugar or fresh fruit) rather than in sugar-sweetened beverages, fruit juices, milkshakes, or sugar-sweetened dairy products. Liquids containing free sugars should be replaced by water or by sugar-free dairy beverages [4].

However, there is an increasingly high consumption of sugars, especially sugar-sweetened beverages (sucrose, high-fruit-content juices, and juice concentrates), although data are more consistent on the consumption of these beverages than on the consumption of other sugars [7]. WHO guidelines on sugar intake for adults and children recommend a reduction in the consumption of free sugars to less than 10% of total energy intake, noting that a reduction to less than 5% generates additional health benefits [7]. It has been reported that children who consume more energy-containing beverages are more likely to be overweight or obese, although this would be attributable not only to the carbohydrates in these drinks but also to the lipids and proteins they contain.

In relation to the dental impact of sugar, the WHO recommends limiting the intake of free sugars to <10% of total energy intake based on moderate quality evidence from observational studies of dental caries, and suggests that a reduction to <5% would further reduce the risk of dental caries (conditional recommendation) [7]. In this regard, added sugars were found to contribute about 14% of daily energy intake in 2- to 9-year-old children in Europe [8].

Besides increasing the risk of overweight/obesity and caries, sugar-sweetened beverages offer a deficient nutrient supply and reduced dietary diversity [9]. However, there have been contradictory reports on their role in the development of childhood obesity. Nine of the thirteen reviews/meta-analyses studied by Keller et al. [10] described a direct association between sugar-sweetened beverages and obesity among children and adolescents [11], while the

remaining four did not. The quality of the studies ranged between poor and moderate, and the two reviews with the highest quality scores showed divergent results. Studies and reviews of optimal quality are needed to fully elucidate this issue [10].

The objective of this study was to determine whether the intake by Spanish schoolchildren of sugars from habitually consumed foods and drinks can be related to overweight and obesity in this population.

## 2. Methods

### Study population

This population derives from a health research project of the Spanish Ministry of Health FEDER-ISCIII PI14/01040. Written informed consent was obtained from parents/guardians of all participants in the study, which was approved by the research ethics committee of the Andalusian Public Health Service. The sample comprised schoolchildren from educational centers in two provinces of Southern Spain (Granada and Malaga). Two age groups were defined in the total sample of 1,000 schoolchildren: a subgroup of children aged 7–8 years and another subgroup aged 9–10 years. Sample size: For an expected obesity prevalence of 10% and precision of 2%, with a 95% confidence interval, the sample size would be 540 (P: 0.10; i: 0.02; α: 0.05). After correction for a finite population: n = 540/ (1+[540/2940]) = 456, the minimum sample size was 456 individuals. Out of the initial sample of 700 individuals. 42 (6%) were excluded for incomplete questionnaire. The drop-outs detected were largely due to missing data on sex or height. The study included 657 schoolchildren (53.5% girls) aged between 7 and 10 years (inclusive). *Methods*: Experienced and specifically trained interviewers administered the questionnaires to participants [12–14]. The encoded questionnaire comprised four sections. Section A gathered data on sex, age, educational center, school year, and life and family habits, among others. Section B was a validated and widely used [15–17] semi-quantitative food frequency questionnaire (FFQ) related to the previous 12 months. It records the consumption or not of each food, the number of times consumed per day, week, or month, and the amount consumed on each occasion in g, mL, or domestic measures (e.g., platefuls, glassfuls, tea/table spoonful, etc.). The daily food and nutrient intake was calculated (in g or mL) from the results by multiplying the standard serving size of each item by the value corresponding to the consumption frequency: never = 0; 1–3 times/ month = 0.07; 1–2 times/ week = 0.21; 3–4 times/ week = 0.50; 5–6 times/ week = 0.80; 1 time/day = 1; and 2–3 times/ day = 2.5 [18,19]. Section C gathered information on anthropometrics and physical activity. Weight (kg) was measured with a floor scale (model SECA 872; Hamburg. Germany) barefoot and in light clothes, height with a stadiometer (model SECA 214; 20–207 cm), and waist circumference (cm) with a measuring tape (model SECA 201), following the CDC Anthropometry Procedures Manual [20]. Participants were then classified as normal weight, overweight, or obese according to the BMI-based classification of Cole et al. [21]. Body fat % was calculated using the equations proposed by Marrodán et al. [22]. Data were also collected on physical activity, including hours of sleep, method of journey to school (walking, car, bicycle, etc.), hours/week of physical education in school, and extracurricular sports activities, gathering all sedentary and non-sedentary activity on each of three non-consecutive days; the results were transformed into METs according to Harrell et al. [23] and Ridley et al. [24]. Carbohydrate and sugar intakes were estimated using the Dial program (Copyright © 2015 Alce Ingeniería) in combination with the AUSNUT 2011–13 food nutrient database, identifying each food from the semi-quantitative FFQ and estimating the amount of each nutrient per 100 g of food [25,26].

Adherence to the Mediterranean diet was estimated by using the KIDMED index [27,28], which contains 16 items, 12 positively scored and 4 negatively scored. Total KIDMED scores

were classified as follows: >8 points = good (optimal Mediterranean diet); 4–7 points = average; and <3 points = poor. Study age ranges were selected according to the FAO/WHO [29].

The occupations of parents were classified according to the 10 groups established by the International Standard Classification of Occupations 2008 (ISCO-08), grouped into low (groups 5–9), medium (groups 3 and 4), and high (groups 1 and 2) levels in accordance with national legislation (1591/2010, 26 November; BOE 17-DIC.2010). *Statistical analysis*: SPSS version 22.0 (IBM. Chicago. IL) was used for the statistical analysis. After a descriptive analysis to calculate means, standard deviations, medians, and maximum and minimum values, t-tests and ANOVA were used for analyses of the data, and stepwise regression and logistic regression analyses were conducted, as specified in table footnotes. $P < 0.05$ was considered significant.

## 3. Results

Table 1 lists the characteristics of the study population, who were between 7 and 10 years of age. The mean (SD) BMI was 18.74 $Kg/m^2$ (SD: 3.17), 10.9% of the children were classified as obese, and the mean energy expenditure (MET) was 8.73 MJ/day (SD: 1.33).

Stepwise regression analysis of the contribution of each food to the total nutrient intake of the children showed that industrial milk shakes were the predominant source of total

**Table 1. Population characteristics.**

|  | Mean | SD | Median | Minimum | Maximum |
|---|---|---|---|---|---|
| Age (yrs) | 9.03 | 0.97 | 9.00 | 7.00 | 10.00 |
| Weight (Kg) | 35.12 | 8.91 | 33.43 | 17.85 | 72.30 |
| Height (m) | 1.36 | 0.08 | 1.36 | 1.12 | 1.69 |
| BMI ($Kg/m^2$) | 18.75 | 3.18 | 18.34 | 11.24 | 31.88 |
| Waist circumference (cm) | 63.59 | 8.53 | 62.00 | 32.10 | 104.00 |
| Waist-to-height ratio) | 0.47 | 0.05 | 0.46 | 0.22 | 0.73 |
| Fat composition* (Fat %) | 24.21 | 5.54 | 23.99 | 4.56 | 49.86 |
| Hours sleeping | 8.66 | 1.09 | 8.00 | 7.30 | 14.00 |
| METs sleeping (MJ/wt/night) | 1.25 | 0.33 | 1.21 | 0.69 | 2.81 |
| Hours walking to school | 0.25 | 0.25 | 0.00 | 0.00 | 0.50 |
| METs MJ journey to school/wt | 0.11 | 0.12 | 0.00 | 0.00 | 0.40 |
| PA hours of physical education | 1.92' | 0.54 | 2.00 | 0.00 | 7.00 |
| METs PA (MJ/wt) | 0.82 | 0.33 | 0.33 | 0.00 | 4.13 |
| Hours of extracurricular PA | 0.46 | 0.33 | 0.43 | 0.00 | 2.14 |
| METs extracurricular PA/day/wt | 0.37 | 0.27 | 0.29 | 0.00 | 1.88 |
| Moderate activity up to 24 h | 12.71 | 1.42 | 13.20 | 6.21 | 15.86 |
| METs sedentary/day/wt | 2.60 | 0.75 | 2.51 | 1.10 | 5.39 |
| Total 24-H METs (MJ/day) | 8.73 | 1.33 | 9.28 | 1.41 | 12.04 |
| Total MJ intake /day | 9.15 | 2.85 | 9.55 | 2.81 | 12.80 |
| % energy fat | 30.76 | 11.63 | 32.70 | 20.90 | 40.70 |
| % energy protein | 12.94 | 7.10 | 13.8 | 5.74 | 20.67 |
| % energy carbohydrate | 54.98 | 9.23 | 56.94 | 7.25 | 71.0 |
| KIDMED | 8.93 | 2.05 | 9.00 | 3.00 | 13.00 |
| % participants in each BMI category |  | Normal weight | Over weight | Obesity |  |
|  |  | 62.3 | 26.8 | 10.9 | p<0.001 |

*Fat % calculated as 106.50 x WHI– 28.36 for boys and as 89.73 x WHI– 15.40 for girls [22].

**MET data for different degrees of activity were based on the values proposed by Harrell et al. [23] and Ridley et al [24].

**Table 2. Stepwise regression analysis of the contribution of each food to the total nutrient intake of the children.**

| Energy (MJ) | $R^2$ | % contribution | Protein | $R^2$ | % contribution | Fat | $R^2$ | % contribution | Carbohydrates | $R^2$ | % contribution |
|---|---|---|---|---|---|---|---|---|---|---|---|
| Milkshake | 0.501 | 50.1 | milk | 0.353 | 35.3 | Milk | 0.336 | 33.6 | milkshake | 0.609 | 60.9 |
| creme caramel | 0.612 | 61.2 | bread | 0.523 | 52.3 | creme caramel | 0.646 | 64.6 | creme caramel | 0.725 | 72.5 |
| Cocoa | 0.682 | 68.2 | milkshake | 0.676 | 67.6 | Milkshake | 0.744 | 74.4 | juice | 0.787 | 78.7 |
| Biscuits | 0.745 | 74.5 | yogurt | 0.787 | 78.7 | Cheese | 0.821 | 82.1 | cereals | 0.830 | 83.0 |
| Bread | 0.805 | 80.5 | buns | 0.851 | 85.1 | Buns | 0.879 | 87.9 | buns | 0.865 | 86.5 |
| Milk | 0.845 | 84.5 | biscuits | 0.885 | 88.5 | Yogurt | 0.928 | 92.8 | bread | 0.901 | 90.1 |
| Buns | 0.885 | 88.5 | salad | 0.909 | 90.9 | Cocoa | 0.979 | 97.9 | biscuits | 0.921 | 92.1 |
| Juice | 0.919 | 91.9 | pasta | 0.932 | 93.2 | | | | milk | 0.948 | 94.8 |
| Yogurt | 0.933 | 93.3 | breakfast cereals | 0.953 | 95.3 | | | | apple | 0.958 | 95.8 |
| breakfast cereals | 0.950 | 95.0 | | | | | | | | | |
| Total sugar | $R^2$ | % contribution | Starch | $R^2$ | % contribution | Added sugars | $R^2$ | % contribution | Free sugars | $R^2$ | % contribution |
| milkshake | 0.747 | 74.7 | breakfast cereals | 0.336 | 33.6 | Milkshake | 0.821 | 82.1 | milkshake | 0.801 | 80.1 |
| creme caramel | 0.843 | 84.3 | bread | 0.550 | 55.0 | creme caramel | 0.921 | 92.1 | creme caramel | 0.902 | 90.2 |
| Juice | 0.911 | 91.1 | biscuits | 0.748 | 74.8 | Juice | 0.979 | 97.9 | juice | 0.979 | 97.9 |
| Apple | 0.932 | 93.2 | buns | 0.845 | 84.5 | | | | softdrink/soda | 0.993 | 99.3 |
| Milk | 0.945 | 94.5 | rice | 0.908 | 90.8 | | | | | | |
| softdrink/soda | 0.954 | 95.4 | creme caramel | 0.941 | 94.1 | | | | | | |
| | | | milkshake | 0.960 | 96.0 | | | | | | |

carbohydrates representing 60.9%, and of total sugars, free sugars, and added sugars, representing 74.7–82.1% of the sugars consumed by participants (Table 2). Statistically significant differences were observed among BMI categories (Table 3) and between the sexes (Table 4) in the intake of energy, total sugars, added sugars, and free sugars. Logistic regression analysis demonstrated a significant association between the maintenance of normal weight and daily activity, measured as 24-h activity or extracurricular sports activity and hours walking to school, and between obesity and the elevated intake of starches, total sugars, free sugars, and added sugars (Table 5).

## 4. Discussion

Obesity and overweight have multiple causes [30]. In this study of schoolchildren in Southern Spain, the intake of all forms of carbohydrates (total carbohydrates, starch, total sugars, free sugars, and added sugars) was calculated from FFQ results, and stepwise regression analysis was used to estimate the contribution of different foods to the intake of each carbohydrate under study. The likelihood of overweight/obesity was associated with a greater daily consumption of energy, total sugars, starch, added sugars, and free sugars, whose reduction is widely recommended to achieve weight control. The WHO has pointed out that nutritional programs have a "double duty", not only to reduce obesity rates but also to address the malnutrition risks posed by an increasing shift from traditional local diets towards a Western diet, usually higher in fat, salt, and sugar and lower in nutritional density [31]. In the present population of schoolchildren, the likelihood of normal weight was significantly greater with lower energy expenditure in sedentary activities, higher energy expenditure in sports activities, and higher total activity/day measured in METs.

**Table 3. ANOVA of the energy and carbohydrate intake (carbohydrates, starch, total sugars, and added sugars) as a function of the BMI of the study population.**

|  | BMI categories | mean | SD | P |
|---|---|---|---|---|
| Total energy intake (MJ) | normal weight | 9.27 | 2.87 | 0.020 |
|  | overweight | 9.25 | 2.79 |  |
|  | obesity | 8.98 | 2.77 |  |
| Carbohydrate (g) | normal weight | 311.16 | 135.94 | 0.028 |
|  | overweight | 310.08 | 133.01 |  |
|  | obesity | 301.02 | 125.09 |  |
| Starch (g) | normal weight | 101.70 | 47.49 | 0.112 |
|  | overweight | 97.47 | 44.96 |  |
|  | obesity | 89.25 | 50.37 |  |
| Total Sugars (g) | normal weight | 175.92 | 107.78 | 0.033 |
|  | overweight | 182.32 | 107.13 |  |
|  | obesity | 158.45 | 96.78 |  |
| Added Sugars (g) | normal weight | 109.85 | 100.29 | 0.046 |
|  | overweight | 119.21 | 100.21 |  |
|  | obesity | 83.90 | 94.10 |  |
| Free Sugars (g) | normal weight | 114.64 | 101.39 | 0.061 |
|  | overweight | 122.60 | 100.29 |  |
|  | obesity | 88.70 | 95.02 |  |
| KIDMED | normal weight | 9.00 | 2.07 | 0.835 |
|  | overweight | 8.91 | 2.18 |  |
|  | obesity | 8.85 | 2.08 |  |

N[a] normal weight = 392; overweight = 170; obesity = 69.

It has previously been reported that the consumption of sugar-sweetened beverages contributes to the development of overweight in children [32]. The intake of industrially-produced milkshakes made the greatest contribution to sugar intake in the present population, representing 60.9% of carbohydrates, 74.7% of total sugars, 82.1% of added sugars, and 80.1% of

**Table 4. Mean comparison analysis (t-test) of energy and carbohydrate intake (carbohydrates, starch, total sugars, and added sugars) between the sexes.**

|  | | Mean | SD | P |
|---|---|---|---|---|
| Energy (MJ) | male | 9.35 | 2.86 | 0.001 |
|  | female | 8.80 | 2.80 |  |
| Carbohydrate (g) | male | 309.36 | 135.89 | 0.002 |
|  | female | 276.27 | 131.67 |  |
| Starch (g) | male | 104.91 | 47.99 | 0.004 |
|  | female | 94.18 | 46.07 |  |
| Sugars (g) | male | 201.02 | 106.16 | 0.009 |
|  | female | 178.63 | 106.57 |  |
| Added Sugars (g) | male | 121.11 | 98.76 | 0.006 |
|  | female | 99.37 | 100.02 |  |
| Free Sugars (g) | male | 125.87 | 99.84 | 0.005 |
|  | female | 103.43 | 100.45 |  |
| KIDMED | male | 8.58 | 1.85 | 0.001 |
|  | female | 9.37 | 2.21 |  |

N[a] male = 295; female = 335.

**Table 5. Logistic regression analysis to determine factors associated with normal weight/overweight+obesity.**

| Reference category is overweight+obesity | OR (95% CI) | P |
|---|---|---|
| KIDMED | 1.44(0.73, 2.16) | 0.270 |
| Ref > median value | | |
| Sex | 1.33(0.74, 1.91) | 0.160 |
| Ref. female | | |
| Median MJ | 1.64(0.97, 3.21) | 0.147 |
| Ref > median value | | |
| Median Carbohydrates | 2.61(1.15, 4.06) | 0.250 |
| Ref > median value | | |
| Median Starch | 0.45(0.27, 0.75) | 0.002 |
| Ref > median value | | |
| Median Sugars | 0.58(0.8, 1.97) | 0.050 |
| Ref > median value | | |
| Median Added Sugar | 1.11 (0.88, 2.78) | 0.040 |
| Ref > median value | | |
| Median Free Sugar | 1.10((0.88,1.37) | 0.040 |
| Ref > median value | | |
| Hours walking to school | 1.63(1.04,2.49) | 0.032 |
| Ref > median value | | |
| Sports/day (METs) | 1.72(1.12, 2.32) | 0.010 |
| Ref > median value | | |
| Sedentary activities /day (METs) | 3.03(1.59, 5.77) | 0.001 |
| Ref > median value | | |
| Total activities/day (24 H) METs | 8.31(4.28, 16.10) | 0.001 |
| Ref > median value | | |

free sugars in their diet. According to these results, it appears especially advisable to replace industrial milkshakes with milk. A finding of particular concern is the very frequent consumption of these beverages instead of fruit or natural juice in the children's mid-morning and afternoon snacks.

In 2015, the Executive Council of the WHO recommended the education of children, parents, and teachers on the importance of consuming healthy food and reducing the intake of sugars and fats; it also proposed the promotion of physical exercise and a reduction in sedentariness [33]. A wide study of Mexican schoolchildren showed that only a small proportion of them ate healthy snacks at school, and it called for greater efforts by educational authorities to promote healthy nutritional behavior and reduce childhood obesity [34]. In general, policies are required to reduce unnecessary sugar intake by babies, children, and adolescents, using fiscal and other measures [35–38] A healthy approach to the consumption of sugar-sweetened beverages and foods should be established during childhood to prevent negative health effects later in life [10]. Products are promoted using a wide range of techniques to reach children through television and the internet as well as in schools, supermarkets, and other settings, and television advertising is known to influences the dietary preferences and patterns of infants and children [39]. Schools can play a key role in campaigns to reduce the excessive consumption of sugary and processed food, and a program to give Maltese schoolchildren bottles of water instead of soft drinks achieved a major reduction in mean energy intake at 12 weeks [40]. The most common sugary and processed food include highly-processed items (e.g., fast foods and snacks), which tend to be low in nutrients (vitamins, minerals and antioxidants) and high in empty calories due to the content of refined flours, sodium and sugar.

Researchers have highlighted the negative effects for children and adolescents of a poorly balanced diet and sedentary activities, including time spent inactive before a screen. The present findings on the relationship of obesity with energy expenditure activity are consistent with strong evidence of an association between screen time and obesity/adiposity and moderate evidence of an association between screen time and higher energy intake [41]. Indeed, particular attention should be paid to increasing physical activity in the present study population, whose sugar intake appears to have a lesser impact, likely attributable to the influence of the traditional Mediterranean diet in our setting. Studies of adolescents have suggested that travelling to school on foot or by bicycle reduces their body fat and is one way of improving the general physical activity of this age group [42].

Some difficulties remain in the definition of sweets, snacks, and foods served in main meals. For instance, researchers in the USA reported problems with the categorization of desserts such as biscuits or ice-cream, which were classified as sweets by some children and snacks by others [43]. There remains a need to establish a consensus on terms and definitions related to sugars and sugar-sweetened beverages, as recently proposed by the European Society for Pediatric Gastroenterology, Hepatology, and Nutrition [4]. The present results are confined to schoolchildren aged 7 to 10 years and do not consider nutritional and life habits that may be acquired in adolescence. A further limitation is that only the intake of sugars and total energy was studied, and the contribution of other energy nutrients (e.g., proteins and fats) was not taken into account.

In conclusion, the reduction of sugar intake, alongside the limitation of total daily intake from energy nutrients such as fats and proteins, is essential to combat the alarming prevalence of obesity among children. Well-designed programs are required that involve parents and schools. Further studies are warranted to verify these findings and examine their implication for public health policies.

## Acknowledgments

The authors thank Layla Davies-Jimenez and Richard Davies for assistance with the English version. This paper will be part of Leticia Heras-Gonzalez's doctoral thesis, being completed as part of the "Nutrition and Food Sciences Program" at the University of Granada Spain.

## Author Contributions

**Conceptualization:** José Antonio Latorre, Fátima Olea-Serrano, Miguel Mariscal-Arcas.

**Data curation:** Leticia Heras-Gonzalez, José Antonio Latorre, Manuel Martinez-Bebia, Nuria Gimenez-Blasi, Fátima Olea-Serrano, Miguel Mariscal-Arcas.

**Formal analysis:** José Antonio Latorre, Manuel Martinez-Bebia, Fátima Olea-Serrano, Miguel Mariscal-Arcas.

**Funding acquisition:** Miguel Mariscal-Arcas.

**Investigation:** José Antonio Latorre, Fátima Olea-Serrano, Miguel Mariscal-Arcas.

**Methodology:** Fátima Olea-Serrano, Miguel Mariscal-Arcas.

**Supervision:** Miguel Mariscal-Arcas.

**Validation:** Miguel Mariscal-Arcas.

**Writing – original draft:** José Antonio Latorre, Manuel Martinez-Bebia, Fátima Olea-Serrano, Miguel Mariscal-Arcas.

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
