## [Decision Letter · Decision Letter 0]

5 Aug 2020

PONE-D-20-10138

Sugar consumption in schoolchildren from southern Spain and influence on the prevalence of obesity

PLOS ONE

Dear Dr. Olea,

Thank you for submitting your manuscript to PLOS ONE. After careful consideration, we feel that it has merit but does not fully meet PLOS ONE’s publication criteria as it currently stands. Therefore, we invite you to submit a revised version of the manuscript that addresses the points raised during the review process.

A number of issues have been identified that need to be clarified and resolved by the authors before considering the possible publication of the manuscript. Authors should carefully answer all the questions indicated by the reviewers, particularly those expressed by reviewer #2. Please note that caution is advised of the possible existence of confirmation bias. It is necessary that the manuscript be rewritten in a more objective fashion and that the analyses and results shown allow the conclusions to be justified by them.

We look forward to receiving your revised manuscript.

Kind regards,

Jose M. Moran

Academic Editor

PLOS ONE

Journal Requirements:

2. Please include additional information regarding the survey or questionnaire used in the study and ensure that you have provided sufficient details that others could replicate the analyses. For instance, if you developed a questionnaire as part of this study and it is not under a copyright more restrictive than CC-BY, please include a copy, in both the original language and English, as Supporting Information. Moreover, please include more details on how the questionnaire was pre-tested, and whether it was validated.

3. In your Methods section, please provide additional information about the participant recruitment method and the demographic details of your participants. Please ensure you have provided sufficient details to replicate the analyses such as: a) the recruitment date range (month and year), b) a description of any inclusion/exclusion criteria that were applied to participant recruitment, c) a table of relevant demographic details, d) a statement as to whether your sample can be considered representative of a larger population, e) a description of how participants were recruited, and f) descriptions of where participants were recruited and where the research took place.

4. We suggest you thoroughly copyedit your manuscript for language usage, spelling, and grammar. If you do not know anyone who can help you do this, you may wish to consider employing a professional scientific editing service.  

Reviewers' comments:

Reviewer's Responses to Questions

**Comments to the Author**

1. Is the manuscript technically sound, and do the data support the conclusions?

Reviewer #1: Yes

Reviewer #2: Partly

Reviewer #3: Partly

2. Has the statistical analysis been performed appropriately and rigorously? 

Reviewer #1: Yes

Reviewer #2: Yes

Reviewer #3: No

3. Have the authors made all data underlying the findings in their manuscript fully available?

Reviewer #1: Yes

Reviewer #2: Yes

Reviewer #3: Yes

4. Is the manuscript presented in an intelligible fashion and written in standard English?

Reviewer #1: Yes

Reviewer #2: Yes

Reviewer #3: Yes

5. Review Comments to the Author

Reviewer #1: Comments to the Authors:

1. Line 123: Methods section: Out of the initial sample of 700 individuals 42 subjects were excluded. That leaves the total sample size of 658. However, author reported that study conducted on 657 schoolchildren. Also, in the tables 3 and 4, the cumulative sample size is 631 and 630, respectively. Were there any drop-out from the subjects during the study period? If there were any drop-outs, I would recommend to mention this either in method or discussion section.

Reviewer #2: The study design can be followed. Critical point: What is new? The results simply confirm some previous studies. Please, make a statement on that point.

Introduction. Several wordings should be corrected. Lines 72f: From the physiological point of view, the cause of weight gain in otherwise healthy people is always (!!) in all ages an imbalance between energy intake and energy consumption; line 77: What is meant with "poor diet"? Please, define; lines 100f: It should be correctly mentioned that the consumption of energy(!)-containing beverages increase the risk to develop an energy imbalance - this is not only related to carbohydrates present in beverages.

Methods. Recruitment: All necessary criteria with respect to sample point selection, number of participants planned, exclusion criteria etc. must be given - is the study representative for Spain? Regression analysis: Why only carbohydrates taken into account? Why not fat intake? According to table 1, fat intake showed a broader variation than the sum of carbohydrate intake!

Discussion. Strengths/weaknesses of the study design must be discussed, eg, the compliance of data collection and the recruitment (see above). Most of the first 2 paragraphs are a repetition of results - can be shortened. What about studies in other European countries like Germany? Eg, KOPS? Line 211f: What is meant with "unhealthy" food? Criteria? And a "poorly balanced diet"? Please, explain.

Conclusion. Since only carbohydrate intake was taken into account, the conclusion is not valid.

Reviewer #3: The authors present an interesting dataset about sugar consumption and the prevalence of obesity in schoolchildren from southern Spain. However, the result is poorly structured and suffers from inappropriate data presentation and analysis as detailed below.

1. The cut-offs for normal weight, overweight, and obesity were not given.

2. The results were barely interpreted:

a. Per Table 1, I would suggest presenting the data by weight status, i.e. underweight, normal weight, overweight, and obesity. Is any participant underweight?

b. How did the authors conclude “industrial milkshakes were the predominant source of total carbohydrates representing 50.9%” (Line 166-168) from Table 2?

c. What are the definitions of total sugars, free sugars, and added sugars? Are they overlapped? What were their effects on body weight, respectively?

d. Did the authors perform analysis of covariance when investigating the association of sugar consumption and obesity?

3. Some sentences are confusing:

Line 64-66: “The likelihood of normal weight was significantly greater with lower energy expenditure in sedentary activities (OR=3.03), higher energy expenditure in sports activities (OR=1.72), and total activity/day measured as METs (OR=8.31).”

Similarly, line 177-178: “Normal weight was found to be associated with physical activity, as measured by METs or total energy expenditure/day, and with energy intake.”

Please rephrase the above sentences to better interpret the data.

6. PLOS authors have the option to publish the peer review history of their article (what does this mean?). If published, this will include your full peer review and any attached files.

Reviewer #1: No

Reviewer #2: No

Reviewer #3: No

---

## [Author Response · Author response to Decision Letter 0]

28 Sep 2020

Comments to the Author

1. Is the manuscript technically sound, and do the data support the conclusions?

Reviewer #1: Yes

Reviewer #2: Partly

Reviewer #3: Partly

2. Has the statistical analysis been performed appropriately and rigorously?

Reviewer #1: Yes

Reviewer #2: Yes

Reviewer #3: No

3. Have the authors made all data underlying the findings in their manuscript fully available?

Reviewer #1: Yes

Reviewer #2: Yes

Reviewer #3: Yes

4. Is the manuscript presented in an intelligible fashion and written in standard English?

Reviewer #1: Yes

Reviewer #2: Yes

Reviewer #3: Yes

5. Review Comments to the Author

\f

Reviewer #1: Comments to the Authors:

1. Line 123: Methods section: Out of the initial sample of 700 individuals 42 subjects were excluded.

That leaves the total sample size of 658. However, author reported that study conducted on 657 schoolchildren. Also, in the tables 3 and 4, the cumulative sample size is 631 and 630, respectively. Were there any drop-out from the subjects during the study period? If there were any drop-outs, I would recommend to mention this either in method or discussion section.

Response: The drop-outs detected by the reviewer were largely due to missing data on sex or height.

---

## [Decision Letter · Decision Letter 1]

19 Oct 2020

PONE-D-20-10138R1

Sugar consumption in schoolchildren from southern Spain and influence on the prevalence of obesity

PLOS ONE

Dear Dr. Olea-Serrano,

Thank you for submitting your manuscript to PLOS ONE. After careful consideration, we feel that it has merit but does not fully meet PLOS ONE’s publication criteria as it currently stands. Therefore, we invite you to submit a revised version of the manuscript that addresses the points raised during the review process.

Thank you for the job done in reviewing your manuscript. Please address any minor issues highlighted by the reviewers before recommending acceptance of the manuscript. 

We look forward to receiving your revised manuscript.

Kind regards,

Jose M. Moran

Academic Editor

PLOS ONE

Reviewers' comments:

Reviewer's Responses to Questions

**Comments to the Author**

1. If the authors have adequately addressed your comments raised in a previous round of review and you feel that this manuscript is now acceptable for publication, you may indicate that here to bypass the “Comments to the Author” section, enter your conflict of interest statement in the “Confidential to Editor” section, and submit your "Accept" recommendation.

Reviewer #1: All comments have been addressed

Reviewer #2: (No Response)

Reviewer #3: (No Response)

2. Is the manuscript technically sound, and do the data support the conclusions?

Reviewer #1: Yes

Reviewer #2: Yes

Reviewer #3: Yes

3. Has the statistical analysis been performed appropriately and rigorously? 

Reviewer #1: Yes

Reviewer #2: Yes

Reviewer #3: Yes

4. Have the authors made all data underlying the findings in their manuscript fully available?

Reviewer #1: Yes

Reviewer #2: Yes

Reviewer #3: Yes

5. Is the manuscript presented in an intelligible fashion and written in standard English?

Reviewer #1: Yes

Reviewer #2: Yes

Reviewer #3: Yes

6. Review Comments to the Author

Reviewer #1: This submitted manuscript evaluated the impact of sugar consumption on prevalence of obesity in school children of Granada and Malaga, Spain. The subject of the paper is interesting that provides new data and the scope covered by the manuscript is worth to be published in PLOS One. The abstract and keywords are fine. The manuscript is well revised and the authors have made appropriate changes as suggested, and thus, I recommend to publish the manuscript in its current form.

Reviewer #2: Obviously, the authors generally answered the questions in their response commentaries adequatly; however, the text was not always changed accordingly. Some examples:

1. Recruitment: The text formulated as response to the reviewers` question should be integrated completely in the Methods section!

2. Line 102: The authors should simply write "energy-containing beverages" and add in brackets (optional) "carbohydrates, fat and protein".

3. Line 77: If a "hyperenergetic diet" is meant, then the authors should use this term! A "poor diet" is not only a question of energy content.

4. Line 215f: The term "unhealthy food" is simply not correct in this context. The study only evaluate "energy intake" with respect to body weight and does not focus on e.g. micronutrients. This sentence must be reworded.

Reviewer #3: The authors did a good job addressing all my questions and concerns except one.

Regarding Table 2 (Line 171-172), the authors interpreted the data incorrectly. R squared (60.9% rather than 50.9%, 74.7-82%) in Stepwise regression analysis doesn't equal to % of the contribution. I would suggest authors add a column in Table 2 as % of contribution and correct the data accordingly in Line 171-172.

7. PLOS authors have the option to publish the peer review history of their article (what does this mean?). If published, this will include your full peer review and any attached files.

Reviewer #1: No

Reviewer #2: No

Reviewer #3: No

---

## [Author Response · Author response to Decision Letter 1]

4 Nov 2020

Reviewer #1: This submitted manuscript evaluated the impact of sugar consumption on prevalence of obesity in school children of Granada and Malaga, Spain. The subject of the paper is interesting that provides new data and the scope covered by the manuscript is worth to be published in PLOS One. The abstract and keywords are fine. The manuscript is well revised and the authors have made appropriate changes as suggested, and thus, I recommend to publish the manuscript in its current form.

Response: The authors are grateful for the positive evaluation of Reviewer 1.

Reviewer #2: Obviously, the authors generally answered the questions in their response commentaries adequatly; however, the text was not always changed accordingly. Some examples:

1. Recruitment: The text formulated as response to the reviewers` question should be integrated completely in the Methods section!

2. Line 102: The authors should simply write "energy-containing beverages" and add in brackets (optional) "carbohydrates, fat and protein".

3. Line 77: If a "hyperenergetic diet" is meant, then the authors should use this term! A "poor diet" is not only a question of energy content.

4. Line 215f: The term "unhealthy food" is simply not correct in this context. The study only evaluate "energy intake" with respect to body weight and does not focus on e.g. micronutrients. This sentence must be reworded.

Response: On point 1, the authors have completely included the text in the Methods section (lines 122-134). In relation to point 2, the authors have changed "sugar-sweetened beverages" to "energy-containing beverages”. Point 3: The authors have changed “poor diet” to “hyperenergetic diet”. Point 4: The authors have changed “unhealthy food” to “sugary and processed food” (lines 219-220 and 222).

Reviewer #3: The authors did a good job addressing all my questions and concerns except one.

Regarding Table 2 (Line 171-172), the authors interpreted the data incorrectly. R squared (60.9% rather than 50.9%, 74.7-82%) in Stepwise regression analysis doesn't equal to % of the contribution. I would suggest authors add a column in Table 2 as % of contribution and correct the data accordingly in Line 171-172.

Response: The authors appreciate the good feedback. The proposed changes to lines 171-172 have been made (lines 177-178 revised manuscript). Also, the authors have updated Table 2 by including the % contribution column.

---

## [Editor Report · Decision Letter 2]

6 Nov 2020

Sugar consumption in schoolchildren from southern Spain and influence on the prevalence of obesity

PONE-D-20-10138R2

Dear Dr. Olea-Serrano,

We’re pleased to inform you that your manuscript has been judged scientifically suitable for publication and will be formally accepted for publication once it meets all outstanding technical requirements.

Kind regards,

Jose M. Moran

Academic Editor

PLOS ONE
---

## [Editor Report · Acceptance letter]

11 Nov 2020

PONE-D-20-10138R2 

Sugar consumption in schoolchildren from southern Spain and influence on the prevalence of obesity 

Dear Dr. Olea-Serrano:

I'm pleased to inform you that your manuscript has been deemed suitable for publication in PLOS ONE. Congratulations! Your manuscript is now with our production department. 

Kind regards, 

on behalf of

Dr. Jose M. Moran 

Academic Editor

PLOS ONE